# Optimal Position of Attachment for Removable Thermoplastic Aligner on the Lower Canine Using Finite Element Analysis

**DOI:** 10.3390/ma13153369

**Published:** 2020-07-29

**Authors:** Won-Hyeon Kim, Kyougnjae Hong, Dohyung Lim, Jong-Ho Lee, Yu Jin Jung, Bongju Kim

**Affiliations:** 1Clinical Translational Research Center for Dental Science, Seoul National University Dental Hospital, Seoul 03080, Korea; wonhyun79@gmail.com; 2Department of Mechanical Engineering, Sejong University, Seoul 05006, Korea; dli349@sejong.ac.kr; 3Seethrough Tech. Corp., Seoul 06149, Korea; dent4u21@naver.com; 4Department of Oral and Maxillofacial Surgery, School of Dentistry, Seoul National University, Seoul 03080, Korea; leejongh@snu.ac.kr; 5Research Center for Advanced Specialty Chemicals, Korea Research Institute of Chemical Technology, Ulsan 44412, Korea

**Keywords:** clear aligner, attachment, optimal positioning, mandibular canine, finite element analysis

## Abstract

Malocclusion is considered as a developmental disorder rather than a disease, and it may be affected by the composition and proportions of masseter muscle fibers. Orthodontics is a specialty of dentistry that deals with diagnosis and care of various irregular bite and/or malocclusion. Recent developments of 3D scanner and 3D printing technology has led to the use of a removable thermoplastic aligner (RTA), which is widely used due to its aesthetic excellence, comfortableness, and time efficiency. However, orthodontics using only an RTA has lower treatment efficacy and accuracy due to the differing movement of teeth from the plan. In order to improve these disadvantages, attachments were used, and biomechanical analyses were performed with and without them. However, there is insufficient research on the movement of teeth and the transfer of load according to the attachment position and shape. Therefore, in our study, we aimed to identify the optimal shape and position of attachments by analyzing various shapes and positions of attachments. Through 3D finite element analysis (FEA), simple tooth shape and mandibular canine shape were extracted in order to construct the orthodontics model which took into account the various shapes and positions of attachments. The optimal shape of a cylinder was derived through the FEA of simple tooth shape and analyzing various positions of attachments on teeth revealed that fixing the attachments at the lingual side of the tooth rather than the buccal side allowed for torque control and an effective movement of the teeth. Therefore, we suggest fixing the attachments at the lingual side rather than the buccal side of the tooth to induce effective movement of teeth in orthodontic treatment with the RTA in case of canine teeth.

## 1. Introduction

Malocclusion is considered as a developmental disorder rather than a disease [1]. A recent study by Isola et al. reported that facial asymmetry and malocclusion was affected by the type and composition of fibers in masticatory muscles [2]. The study also reported that the composition of the fiber types of masticatory muscle are related to the vertical overlap of the anterior teeth in centric occlusion [2]. However, the cause of malocclusion has not been clearly identified yet [1]. Orthodontic treatment is a type of dental treatment that deals with diagnosis and care of various irregular bite [1] and/or malocclusion. Conventionally, fixed appliances were widely used to treat various irregular bites and/or malocclusion. These appliances manually manufactured during the treatment process are produced through dental technicians, requiring a lot of time for planed movement of teeth during each orthodontic step [3]. As such, experienced dental technicians and orthodontists are essential in orthodontic treatments [3].

As an improvement of the conventional orthodontic treatment, removable thermoplastic aligner (RTA) was developed based on CAD (computer-aided design)/CAE (Computer-aided engineering), and in the treatment using an RTA, the orthodontic treatment device can be produced after establishing virtual orthodontic stages using the CAD software [4]. Previous studies reported that the number of patients seeking treatment with an RTA is increasing [5] because it is more comfortable and aesthetic compared to the conventional fixed appliances [6]. Buschang et al. [7] have compared RTAs and fixed appliances with respect to time efficiency, and found that treatment with the RTA required significantly less visits, less emergency visits, less emergency chair time, and lowest total chair time than fixed appliances [7]. In addition, one study has reported that the RTA with periodic loading allowed the resorbed cementum to prevent and heal the root resorption [8]. Eissa et al. [9] have found that root resorption in the RTA was significantly lower than the conventional fixed appliances. However, an RTA shows a different orthodontic force compared to arch wire and brackets due to rebound force, making orthodontic treatment difficult in severe malocclusions because it cannot control the movement of the teeth as the orthodontist intended [10,11]. Therefore, in a treatment process using RTA alone, correct orthodontic treatment is not achieved as it is difficult to predict the movement of teeth in the course of the treatment.

Previous studies have reported that auxiliary devices such as attachments and interarch elastics are required to improve the predictability of teeth movement during the orthodontic treatment [6]. Previous research related with attachment and RTA were conducted on the use of attachments as well as the thickness and shape of the attachments in the movement of specific teeth. Previous research has reported that attaching horizontal rectangular attachment with a thickness of 1 mm on the buccal and lingual sides helps the extrusion and rotation of teeth [12,13]. Also, it was reported that RTA retention was improved when oval-shaped attachments were attached on teeth [14].

However, in the orthodontic treatment, various movement mechanisms such as extrusion, intrusion, rotation, and torque are applied [15], and the positions of the attachments attached on the surface of the teeth vary. In the previous studies, most of the analyses were made on the effectiveness of attachments at specific orthodontic treatment stages, rather than on the movement mechanism and the position of attachments [6,12,13,14]. Therefore, there is a need for a comparative analysis on the various positions of attachments fixed on the surface of teeth and the movement mechanism of teeth at each orthodontic stage, as well as research on the distribution of stress delivered to the inside of teeth.

For several decades, 3D simulation analysis was widely used in the field of dental research by building a hypothetical 3D finite element model assuming the dental treatments and surgical conditions [6,12,13,14,16,17,18]. Specifically, the finite element analysis (FEA) is non-invasive and is a virtual model, it has the advantage of being able to predict the results without direct application in patients [6,12,13,14,16,17,18]. In addition, it allows analysis to be conducted by simulating the procedural method and environment that are difficult to apply in actual clinical setting.

Therefore, the objective of our research was to simulate various shapes of attachments for each of the four orthodontic treatment situations (extrusion, intrusion, rotation, and torque) using the FEA to derive the optimum shape of attachment for each situation, and to analyze the best position for attachments by simulating various attachment positions for each orthodontic treatment situation using the derived attachment shapes. The hypotheses of our study are as follows: (1) the contact area between the RTA and the attachment will be larger when the round type is used than the square type, and (2) the movement of teeth will be more effective when the attachment position is closer to the apical end.

## 2. Materials and Methods

### 2.1. Designing Attachments for Various Orthodontic Treatment Situations

Various shapes of attachments were designed using the program Solidworks (Solidworks 2016, Dassault Systemes SolidWorks Corp., Waltham, MA, USA) for each of the four movements (extrusion, intrusion, rotation, and torque) that occur during orthodontic treatment process. The shapes of attachments designed for extrusion and intrusion included foursquare, half round at a cross-section, half circle at a longitudinal section, half triangle, and oblique, with dimensions 1 mm × 1 mm × 0.85 mm (length × height × thickness) (Figure 1a–j). The shapes of attachments for extrusion and intrusion are identical but the attachments were applied according to the direction of teeth movement as the teeth move in the opposite directions (Figure 1a–j). The shape of attachment for rotation had a plane perpendicular to the direction of rotation so that the two moment of the tooth was applied, and the surfaces to which load was applied were classified to have an angle of 90 degrees, 65 degrees, and 45 degrees to the attachment surface of the teeth (Figure 1k–m). The dimensions for the attachments for rotation were 0.5 mm × 0.75 mm × 0.25 mm (length × height × thickness). For the attachments for torque, four shapes having half round, half round at a cross-section, half round at a cross-and longitudinal-sections, and 45 degrees were designed so that the teeth can move sideways, with dimensions 0.5 mm × 1.2 mm × 0.5 mm (length × height × thickness) (Figure 1n–q).

### 2.2. Finite Element Analysis for Deriving the Optimum Shape of Attachments for Each Orthodontic Treatment Situations

Bone and teeth shapes were designed to analyze the optimal shape of various attachments assuming each orthodontic treatment situation (Figure 2a). The shapes of bone and teeth applied to all attachments were applied identically, and the shapes of bone and teeth were simplified instead of being extracted from the cone beam computed tomography (CBCT) from actual patients (Figure 2a).

The bones were classified into cortical and cancellous bones, and the teeth part were constructed as the teeth and the periodontal ligament (PDL) (Figure 2a,b). By referring to previous studies, the thicknesses of the cortical bone and the PDL were set to 2 mm and 0.2 mm, respectively (Figure 2a,b) [19,20,21]. RTA was designed for each orthodontic treatment situation, and RTA was implemented with thickness of 0.3 mm based on previous literature (Figure 2b) [19]. The material properties of each component except for PDL were applied with reference to previous literature (Table 1) [18,19,22,23].

In previous studies, a comparative analysis between experimental data and hyperelastic properties was performed in order to derive the material properties of PDL. In our study, the Ogden model was used with considerations for the hyperelastic properties for the material properties of PDL and the variables applied are presented in Table 2 [24].

For the 3D CAD models with considerations for each orthodontic treatment situation, the FEA program ABAQUS (ABAQUS CAE2016, Dassault systems, Vélizy-Villacoublay, Yvelines, France) was used to form the hexahedral element (C3D8R) for all components except for the attachments, whereas attachments were formed using tetrahedral element (C3D4). The mesh size for teeth, PDL, RTA and attachments ranged from the minimum of 0.05 mm to the maximum of 0.15 mm, whereas the bone ranged from the minimum of 0.15 mm to the maximum of 0.8 mm. Both the minimum and the maximum values for attachment was set to 0.05 mm. The number of elements and nodes for each model was 43,098 and 37,790, 311,954 and 298,148, and 39,005 and 30,992 for the cortical bone, cancellous bone and the PDL, respectively, and these values were applied identically to all models. In contrast, the elements and nodes for teeth, RTA and attachments varied by each tooth movements (extrusion, intrusion, torque, and rotation), as the range of element was from 140,324 to 174,216 for teeth, from 24,640 to 36,728 for RTA, and from 6412 to 7938 for attachments. In addition, the range of nodes was from 132,770 to 164,890 for teeth, from 17,816 to 26,452 for RTA and from 5434 to 6800 for attachments.

In the four orthodontic treatment models, “tie contact” was applied assuming complete union and bonding state among bone, teeth, PDL and attachments, whereas a friction coefficient of 0.2 was applied by assuming a sliding state for the surface of contact among RTA, teeth and attachment [23]. The four orthodontic treatment models completely controlled the side and bottom of the bones so that movement and rotation did not occur in all directions (Figure 3a–d).

As the movement and rotation of teeth are generated by the RTA, load was applied by controlling the displacement and rotation of RTA. The orthodontic treatment models for extrusion, intrusion and torque applied load by the means of displacement control, whereas load was applied in the rotation orthodontic treatment model using rotation control method (Figure 3a–d). The extrusion model was moved 0.05 mm in the positive direction on the y-axis while the intrusion model was moved in the negative direction. The torque model was moved 0.05 mm in the positive direction of the x-axis, while 1 degree rotation was applied in the negative direction based on the y-axis for the rotation model (Figure 3a–d) [3].

The FEA used in this study built a virtual 3D simulation model which allows to predict the results by applying realistic load and boundary conditions. Also, to develop a medical device applicable in human body in the dental and medical field, FEA is widely used to compare according to design and material properties because it can be reproduced by building a virtual surgical environment [16,17,18,20,21,25,26,27]. Moreover, for comparison by design of the medical device, the rest of the parts except the device design was designed identically, and it was built in a simple form for comparative analysis [28,29,30,31]. Therefore, our study performed FEA considering the simple shape of teeth and bones to derive the optimal shape from various attachment designs.

After applying the load, the Peak von Mises stress (PVMS) generated in the bone, teeth, PDL, RTA and the attachments, and the contact pressure generated in the attachments were measured for comparative analysis according to the shape of attachment by each orthodontic application. In addition, the teeth were divided into upper and lower areas with the PDL as the reference to measure the maximum displacement and rotation in each area (Figure 4). For the rotation value of attachment, a line was drawn from the buccal side to the lingual side connecting the points of buccal and lingual sides on the tooth at the y-axis to measure the change in the angle after the load was applied. Once load was applied, the contact rate was analyzed according to attachment shape by measuring the contact area of the attachment on the surface of contact between RTA and attachment.

### 2.3. Constructing a Finite Element Model of the Mandibular Canine Considering Various Attachment Positions for Each Orthodontic Situation

After deriving the optimal shapes of attachments for each orthodontic treatment situation based on the results of the FEA of simplified orthodontic treatment model, a 3D finite element model was simulated using the mandibular CBCT image used in previous studies (Figure 5a) [20,21]. The study results of Asiry et al. [32] showed highest prevalence of molars and canines. Also, Kravitz et al. [33] reported that in the teeth movement using only RTA, the extrusion, intrusion and rotation of mandibular canine was less accurate than maxilla canine. A previous study also performed FEA considering a single tooth model of canine [19]. Therefore, our study built a mandibular canine model and performed FEA. Based on previous studies, the thicknesses of the cortical bone, mucosa and PDL were set to 2 mm, 2 mm, and 0.2 mm, respectively [22,23,26,27] (Figure 5b,c). A single tooth shape was considered to compare the displacement and rotation according to the position of attachments, and the mandibular canine was extracted from the mandibular CBCT image (Figure 5c).

In order to compare the biomechanical properties of teeth according to the position of attachments attached on teeth, a four attachment positions were considered for extrusion, intrusion and torque attachments as shown in Figure 6a–c. For rotation attachments, a nine attachment positions were simulated by combining two attachments at each lingual and buccal sides of the tooth, respectively (Figure 6d). Previous literature was referenced for the material properties of the cortical bone, cancellous bone, mucosa, PDL, the teeth and the attachments, and the applied parameters are presented in Table 1 and Table 2 [20,22,23,24,25]. All components were formed using the tetrahedral element (C3D4). The mesh size of the cortical bone, cancellous bone, mucosa and the teeth were set to range between the minimum of 0.15 mm and the maximum of 0.6 mm, whereas the minimum and the maximum of the PDL and attachments were set to 0.05 and 0.15 mm, respectively. The number of elements and nodes for the cortical bone, cancellous bone, mucosa and the teeth were identically applied as 2,201,481 and 184,561 in all tooth movements, respectively. In contrast, the number of elements and nodes for the attachment varied depending on the shape—the number of elements ranged from the minimum of 156,754 to the maximum of 488,890 while the number of nodes ranged from the minimum of 49,569 and the maximum of 144,822.

### 2.4. Contact, Boundary and Loading Conditions of Mandibular Canine Treatment Models

All FEA models by tooth movements and attachment positions were all completely constrained so that movement and rotation did not occur in either side of the elements of cortical bone, cancellous bone and the mucosa (Figure 7a). As the bone and the mucosa, bone and PDL as well as the PDL and the tooth are connected as one, the nodes between neighboring elements were combined using the “tie contact” condition, whereas the combination condition between the teeth and the attachment assumed attachment by resin. In our study, the contact area occurred to the attachments by the orthodontic device was identified based on the results analyzed through the simple teeth finite element model. Load was applied based on the contact surface applied to each attachment (Figure 6b). In previous studies, a measurement sensor was attached to teeth model to measure the load that resulted in the teeth when the orthodontic device moved 0.5 mm. These previous studies were referenced and a load of 11.821N that occurs in the teeth for 0.5 mm displacement was applied on the contact surface of the attachment (Figure 7b) [34].

In order to derive the optimal position for each attachment, the PVMS values in the attachment as well as the teeth and the values of teeth displacement (mm) and rotation (degrees) were compared and analyzed after load was applied to the attachment.

This study used FEA to build various models according to design and measure stress distribution and displacement for comparative analysis. As the FEA built the rest of the parts identically except the device design, it is possible to compare and analyze the stress distribution and displacement values according to change in design [6,12,13,14,16,17]. Therefore, this study derived and compared the single value of each model and did not use statistical analysis.

## 3. Results

### 3.1. The Result for Various Shapes of Attachment in Four Orthodontic Treatment Situations

In the results of the FEA of attachments for extrusion, intrusion, rotation, and torque, the PVMS values, contact pressure, and contact areas were analyzed and the displacement values in the upper and lower regions of the teeth were measured.

In the extrusion case, the PVMS values of the bone and PDL from the EX1 to EX4 showed similar values and EX5 showed the lowest measurement. The PVMS values of teeth in the EX4 was measured as the highest whereas the lowest stress was shown in the EX3. The contact pressure and PVMS values of the RTA and attachments in the EX3 was measured as the lowest while the highest stress was measured in the EX5 (Table 3) (Figure 8a–e). In the intrusion case, the PVMS values of the bone, teeth, PDL and attachments did not show differences by group (Figure 8f–j), and EX5 showed the highest stress for the PVMS value of the RTA. The contact pressure of the EX3 was measured to be lower compared to other groups as in the case for extrusion (Table 3). Although rotation did not show a difference among all groups for the PVMS values of bone and PDL, the PVMS values of teeth and attachments as well as the contact pressure of attachments were the lowest for RO2. In contrast, the PVMS value of the RTA in the RO3 was measured to be the lowest (Table 3) (Figure 8k–m). In the case of torque, the PVMS values of the bone and PDL were measured to be lower in TO1 and TO2 compared to that of TO3 and TO4, and the stress in teeth was at least twice as higher for TO1 and TO2 (Table 3). The PVMS values of RTA and attachments were lower for TO1 and TO3 compared to the rest of the two groups (Figure 8n–q), while the contact pressure of attachments was measured to be very high in TO4 compared to the rest of the groups (Table 3).

In extrusion case, the upper and lower displacement values of teeth were measured to be the lowest in the EX5 and the rest of the groups showed similar results (Table 4). The contact area of the RTA and attachments in the EX3 showed the highest contact area compared to other groups (Table 4) (Figure 8a–e). In intrusion case, the IN3 showed the highest contact area (Figure 8f–j) and teeth displacement values did not show a difference among the groups (Table 4). In rotation case, the RO3 showed the lowest rotation value in the upper and lower teeth and the RO2 showed the highest rotation (Table 4). The contact area of attachment was shown to be in the order of RO1, RO2, and RO3 but there was no difference among the three groups (Figure 8k–m). In the case of torque, the displacement values in the lower region of the teeth did not show difference among the groups, but the TO1 and the TO2 showed a higher displacement compared to the TO3 and the TO4 in the upper region of the teeth (Table 4). In contrast, the contact area of attachment in the TO1 and the TO2 was found to be 28.42% and 31.93%, respectively, which was more than 15% lower than that for the TO3 and the TO4 (Figure 8n–q).

### 3.2. Results of Finite Element Analysis for Four Teeth Movements to Derive Optimal Attachment Position

In the results of the FEA for four attachment types, the PVMS values for the cortical bone, cancellous bone, mucosa and the PDL did not show any difference among the groups, but the PVMS values of the attachment and the teeth showed a difference according to attachment positions.

The PVMS and displacement values of attachments and the teeth are shown in Figure 9a–g and the rotation values of rotation attachment were shown in Figure 8h after measuring the angle before and after loading by extracting the nodes from the center of the buccal and lingual sides of the surface of attachment.

In the extrusion attachment model, it was confirmed that the PVMS values of attachments and the teeth decreased as the attachment position moved from the incisal to the gingival positions of the teeth at the buccal side whereas the displacement values of the teeth increased as opposed to the stress distribution patterns (Figure 9a,e). In addition, the results were similar for the lower region of the lingual side and the lower region of the buccal side of the teeth (Figure 9a,e). In the intrusion attachment model, unlike in the extrusion model, the PVMS values increased as the attachment position moved from incisal to gingival positions. The PVMS values were the lowest when the attachments were fixed at the center of the teeth in the buccal side and the highest when attached at the center of the lingual side of the teeth (Figure 9b). The displacement values of attachment and teeth were the highest when the attachment position was at the center of the teeth in the buccal side unlike in the extrusion attachment, and the lowest displacement value was observed when the attachment was positioned at the center of the lingual side of the teeth (Figure 9f). In the torque attachment model, there was no difference in the PVMS of the teeth values when the attachments were the teeth at the buccal side and the lowest PVMS of the teeth was observed when the attachment was fixed on the upper region of the lingual side of the teeth (Figure 9c). The PVMS values of attachments were the highest when the attachments were on the lower region of the teeth (Figure 9c). In the buccal side of the teeth, the displacement values of the attachment and teeth tended to decrease as the position moved to downward, and the displacement values for upper region of the lingual side of the teeth were similar to those of the upper region of the teeth at the buccal side (Figure 9g).

In the rotation attachment model, the PVMS values of the teeth did not show a difference among the groups but the PVMS values of attachment showed a different among the groups. The PVMS values of attachments were measured to be the lowest in the ROUD and ROMD models, followed by the ROUU and ROMU models (Figure 6d). In the rest of the models excluding the four models, a high value of at least 120 MPa was observed (Figure 9d). The rotation for the rotation attachment ranged from the minimum of 0.39 degrees to the maximum of 0.58, in the order of RODD, RODU, RODM, ROMD, ROUD, ROMM, ROMU, ROUM, and ROUU from the highest to the lowest (Figure 9h).

## 4. Discussion

In orthodontics, it is considered that the treatment method using RTA cannot deliver adequate load for effective and stable movement of teeth [3,4,19,23]. Previous studies compared rotation and movement of teeth with or without attachment [3,4,19,23], and Barone et al. [3] reported that in incisors and canines, each rotation of teeth was measured as 0.48 and 0.16 degrees without attachment and 0.56 and 0.37 degrees with attachment. A study by Savignano et al. [4] showed 37.4 and 42.8 mm of teeth movement each in vertical and horizontal attachment, and it was measured as 26.7 mm without attachment. Gomez et al. [19] confirmed 0.024 mm of teeth movement with only RTA, 0.175 mm with attachment, and the inclination of teeth was not controlled without attachment. In addition, the use of RTA only may result in the movement that was not intended by the orthodontist. For these reasons, previous studies conducted biomechanical assessment and analysis on RTA and attachments using the FEA, which confirmed that a more effective movement of teeth was induced by attachments [3,4,19,23]. However, the shape of attachment and the position on the teeth are determined by the orthodontist as well as the dental technician, and the analysis on the shape and position of attachment on the teeth according to orthodontic treatment situations was inadequate. Therefore, in the present study, we conducted the FEA to derive the optimal shape and attachment positions for effective teeth movement when performing orthodontic treatments using the RTA.

In our study, the attachments for extrusion, intrusion, torque, and rotation were simulated considering the four types of teeth movement, and three to five shapes were designed in order to derive the optimal shape for four teeth movements. For the attachments of the extrusion, intrusion and torque, load was applied using displacement control, and 0.05 mm was applied. We tried to use the loading condition of 0.15 mm which was used in previous studies [19], but the analysis did not converge due to excessive deformation of the PDL in the extrusion attachment model. The reason for this is the material properties of PDL reported in the previous literature use stress–strain curve values, whereas, our study was applied in the hyperelastic properties were considered using the Ogden model [19,23]. Therefore, the stresses and strains generated in the PDL have been shown to be different between previous and our studies. In previous studies that performed the FEA of orthodontics, rotation values of 1 degree or less were applied to analyze the rotation of the teeth [3,4]. In our study, the teeth are rotated step by step during orthodontic treatment and a rotation value of 1 degree was applied. In the results of analysis for deriving the optimal shape of attachments, it was confirmed that the stress distribution in the cortical bone, cancellous bone and PDL did not show a difference according to the changes in shapes. In the comparison of the PVMS for extrusion attachment, we confirmed that stress was concentrated at corners of attachments that receive load by RTA and the surfaces that attach to the teeth (Figure 8a–e). In particular, the attachment of the EX5 showed more than twice the PVMS and contact pressure compared to other groups and it is predicted that the attachment is likely to high failure risk or detachment than other shapes. In contrast, the attachment of the EX3 showed the lowest values of the PVMS and contact pressure than other shapes (Figure 8c). In addition, the nodes in the contact area between the orthodontic device and the attachment were extracted and compared after dividing the sum of contact surface of each node by the total area in order to identify the contact area between the attachment and the orthodontic device. The contact area was the highest for the cylinder-shaped EX3 and the lowest for the EX4 (Figure 8a–e). In the results for intrusion attachments, the contact pressure of attachments was twice as lower in the EX3 compared to other shapes while the contact area was the highest (Figure 8f–j). However, there was no difference in the PVMS values of the bone, teeth, PDL, RTA, and attachments (Table 3). Therefore, the cylinder-shaped EX3 was predicted to be most appropriate shape for extrusion and intrusion attachments due to the lower risk of fracture and desirable stress distribution. In rotation attachment models, the PVMS value of the RTA was measured to be the lowest for the RO3 but the PVMS values of the attachment and the teeth, and contact pressure of the attachment were the lowest for RO2 (Figure 8k–m). Contact area did not show a difference among the three shapes, but the rotation of teeth was measured to be the highest in RO2 (Table 3). Therefore, the RO2 was predicted to be most appropriate shape for rotation attachments. In the attachment for torque, the PVMS values of the TO1 and TO2 was measured to be more than twice as high compared to the other shapes. Additionally, the contact area of the TO1 and TO2 was showed lower than other shapes (Figure 8n–q). In the attachment of the TO4, the contact pressure was highest due to the stress concentration in the lower region of triangular shape (Figure 8q). The shape of TO3 showed desirable stress distribution compared to other shapes with the highest contact area of the attachment (Figure 8p). Through the results of our study, we confirmed that the distribution of stress delivered to the attachment from the orthodontic device or contact area were high when the attachments were produced in cylinder shape.

After deriving the optimal shape of attachment for each orthodontic situation, the FEA of orthodontic treatment were performed considering various attachment positions. In the case of extrusion, the PVMS values of the teeth and the attachment was the lowest when the attachment was positioned at the lower region of the buccal side of the teeth, and the teeth movement was also higher compared to other attachment positions. Unlike extrusion, the PVMS values of the attachment and the teeth was the lowest when the attachment was positioned at the center region of the buccal side of the teeth in intrusion. Although the movements of attachment and teeth was the highest in the lower region, those of the central region were similar to those of the lower region. In buccal sides of teeth, we predict that attachments will be most effective when they were positioned at the lower region of the anterior of the teeth in case of extrusion, and at the center of the anterior of the teeth in case of intrusion.

However, for both extrusion and intrusion, it seems that unintended movement will result along with rotation of the teeth when the attachment is position at the buccal side of the teeth (Figure 10a–c,e–g). In contrast, when the attachments were positioned at the lingual side of the teeth, there was almost no rotation of tooth and it was confirmed that the teeth moved as planned (Figure 10d,h). In the case of the torque, the PVMS values of the teeth and the attachment was the lowest when the attachment was positioned at the upper region of the buccal side of the teeth, and the movements of the teeth and the attachment was identified to be the highest. However, as with extrusion and intrusion, it was confirmed that teeth rotation was controlled and planned movements were observed when the attachments were attached on the lingual side of the teeth rather than the buccal side of the teeth (Figure 10h,l). A previous study also reported that torque inconsistency occurred at the initial treatment stage when the orthodontic device was applied to the lingual side compared to the buccal side, and the vertical position of the teeth affected the lingual control of torque by the orthodontic device [35]. Therefore, considering teeth movement, it would be efficient to perform orthodontic treatment by fixing the attachments at the lingual side of the teeth or at both lingual and buccal sides of the teeth for torque control. For rotation attachments, the rotation value was 0.5 degrees higher regardless buccal side positions of the teeth when attachment was fixed at the lower region of the lingual side of the teeth, and the rotation value was higher than 0.5 degrees regardless of the position at the lingual side of the teeth when the attachment was fixed at the lower region of the buccal side of the teeth (Figure 9h). However, when attachments were fixed at the lower region of the buccal side of the teeth, there was high stress in the attachment regardless of the attachment position at the lingual side of the teeth, and the lowest stress was observed for the lower region of the lingual side of the teeth and the upper region as well as the center of the buccal side of the teeth (Figure 8d). Therefore, we predict that the stress distribution of the attachment and teeth movement are the most effective for rotation attachments between ROUD and ROMD (Figure 9h). It seems that even in rotation attachments, fixing the attachment at the lingual side of the teeth greatly affects teeth movement and stress distribution in attachments.

We derived the proper attachment positions at the buccal side of the canine teeth through the results of our study but also confirmed that the attachment position at the lingual side of canine teeth rather than the buccal side has high correlation with canine teeth movement and torque control. Therefore, in order to control canine teeth movement as intended by the orthodontist and to control torque in orthodontic treatment using RTA, it may be helpful to place the attachments at the lingual side of the teeth or use both the buccal side and the lingual side of the canine teeth.

However, our study was difficult to fully reflect the actual orthodontic treatment setting because it analyzed tooth movement and rotation for a single canine tooth. The FEA in our study applied the load using the contact area occurring on the attachment. In addition, our study performed biomechanical analysis of while wearing orthodontic appliances and did not consider the conditions that may occur from factors such as masticatory movements and masseter muscles after wearing orthodontic appliances. Isola et al. reported that there is a correlation between the type and property of masseter muscles, muscle activity and facial asymmetry, and malocclusion [2]. Therefore, future pre-clinical and biomechanical studies with cadaveric bone are required for the efficacy of multiple teeth movement and pre-clinical evaluation assuming actual orthodontic treatment conditions. Moreover, additional research is required for more effective orthodontic treatment through analysis between masseter muscle fiber characteristics and malocclusion after wearing orthodontic appliances.

## 5. Conclusions

Through our study, we confirmed that desirable stress distribution was observed by inducing a high contact area between the attachment and the orthodontic device when a cylinder-shaped attachment was used on the removable thermoplastic aligner used in orthodontic treatment. In addition, we confirmed that torque control and intended movement were achieved when the attachments were positioned at the lingual side rather than the buccal side of the canine teeth. Therefore, the attachment used in the removable thermoplastic aligner treatment of canine teeth is recommended to attach a cylinder form to the lingual side of the canine tooth. In intrusion, it is considered that attachments are placed at the buccal side of the canine tooth as well as placing them at the lingual side of the canine tooth reduces the movement of the canine. However, as our study only considered a single tooth of canine, further studies are required to consider biomechanical analysis with related to multiple numbers of teeth and analysis between the masticatory muscle and occlusion force that affect malocclusion [2].

## Figures and Tables

**Figure 1 materials-13-03369-f001:**
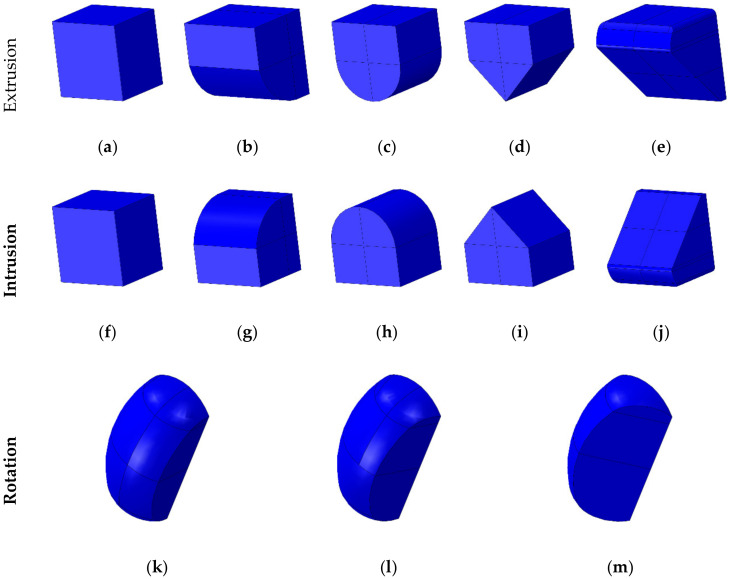
The various design of attachment for four orthodontic treatment situations. The shape of each attachment is defined as follows: (**a**) EX1; (**b**) EX2; (**c**) EX3; (**d**) EX4; and (**e**) EX5 for extrusion attachment; (**f**) IN1; (**g**) IN2; (**h**) IN3; (**i**) IN4; (**j**) IN5 for intrusion attachment; (**k**) RO1; (**l**) RO2; (**m**) RO3 for rotation attachment; (**n**) TO1; (**o**) TO2; (**p**) TO3; and (**q**) TO4 for torque attachment.

**Figure 2 materials-13-03369-f002:**
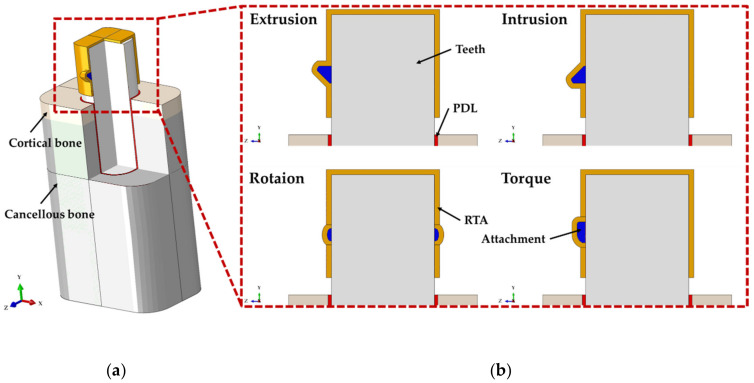
Finite element model built for each of the 4 orthodontic treatment situations. (**a**) Assembled model with bones, teeth, periodontal ligament (PDL), attachment, and removable thermoplastic aligner (RTA). (**b**) Extrusion, intrusion, rotation, and torque situations.

**Figure 3 materials-13-03369-f003:**
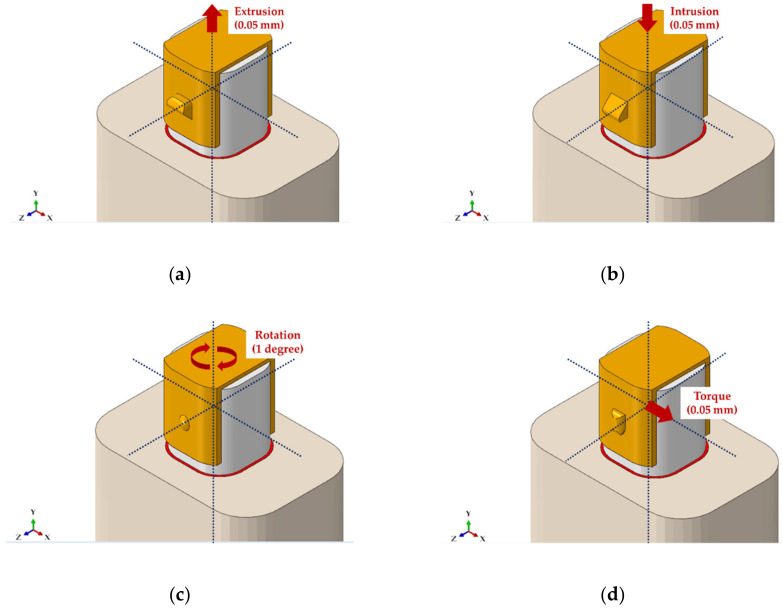
Loading conditions for treatment models by orthodontic situation. (**a**) Extrusion attachment; (**b**) Intrusion attachment; (**c**) Rotation attachment; and (**d**) Torque attachment.

**Figure 4 materials-13-03369-f004:**
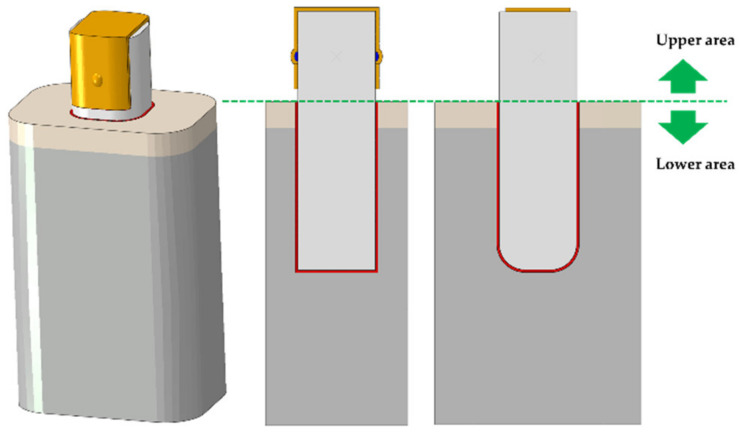
The teeth were separated into upper and lower regions based on the coronal position of PDL in order to analyze the displacement and rotation values of teeth according to various designs.

**Figure 5 materials-13-03369-f005:**
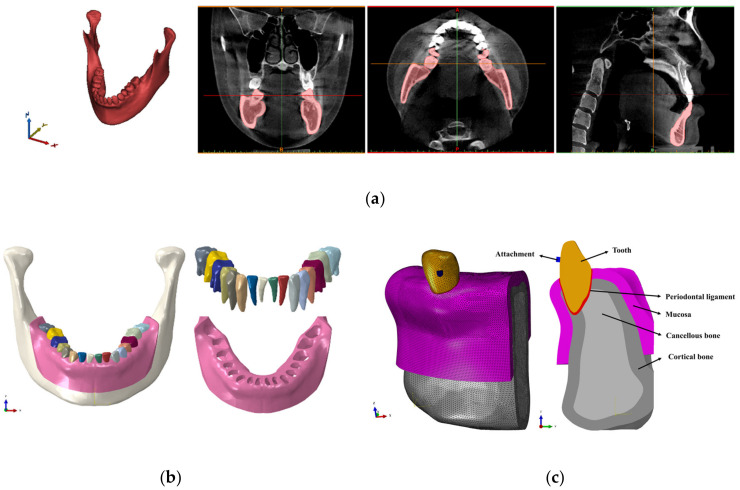
The process of building a 3D mandibular finite element model that includes the teeth, mucosa, PDL, cortical bone, cancellous bone and attachments and the cross-sectional view of the mandibular canine orthodontic treatment model. (**a**) Mandibular reconstruction process through 3D CBCT; (**b**) Building cortical bone, cancellous bone, mucosa and teeth models; and (**c**) Building orthodontic treatment model with attachments in place.

**Figure 6 materials-13-03369-f006:**
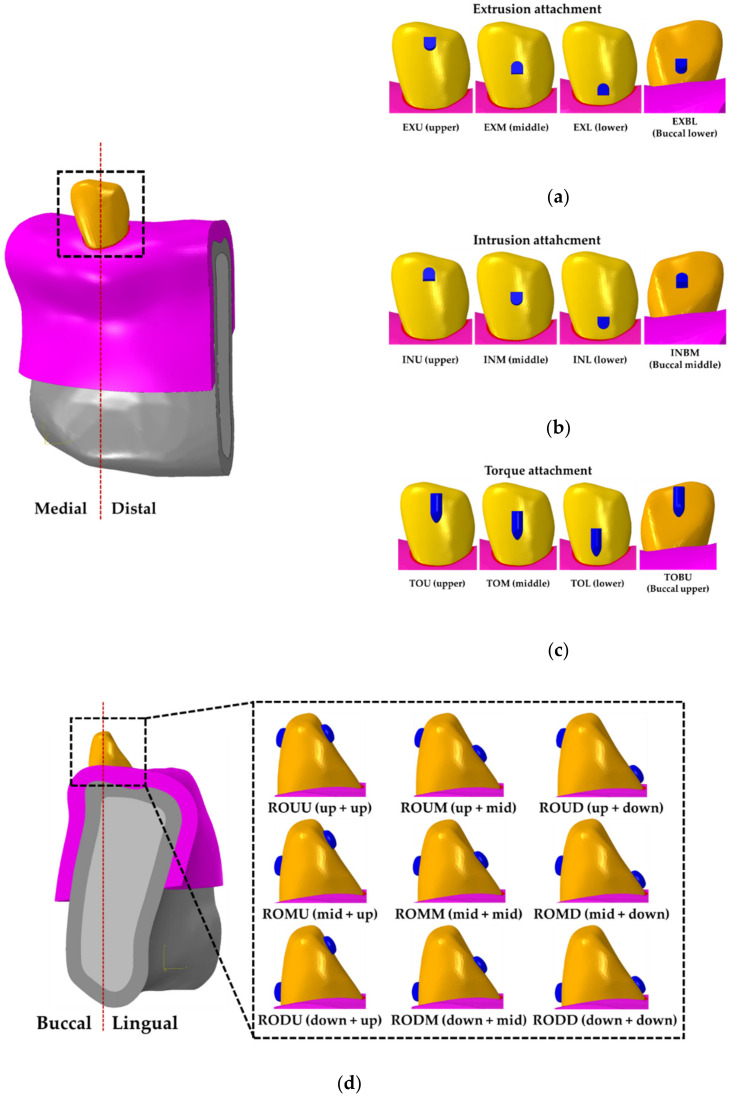
Construction of the treatment model considering various positions of the attachment on teeth by four teeth movements. (**a**) Extrusion attachment; (**b**) Intrusion attachment; (**c**) Torque attachment; and (**d**) Rotation attachment.

**Figure 7 materials-13-03369-f007:**
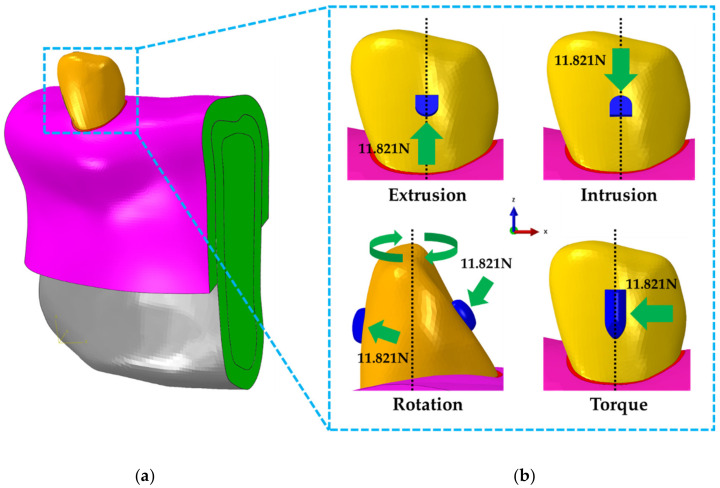
Boundary and loading conditions for orthodontic treatment models considering 4 types of attachments. (**a**) Both sides of the bone model (Green) were fixed in all directions; (**b**) For extrusion and intrusion, 11.821N was applied in the direction of the z-axis on the contact surfaces of the attachments at the gingival and incisal sides. For torque attachments, −11.821N was applied in the direction of x-axis. In addition, as for the rotation attachments, pure moment was applied by applying 11.821N perpendicularly to the plane cut 65 degrees to the attachment at the buccal and lingual sides.

**Figure 8 materials-13-03369-f008:**
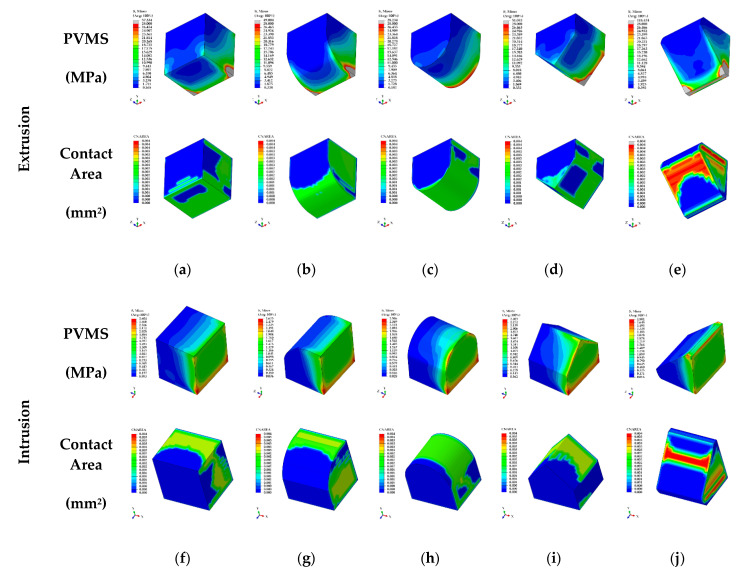
Results of stress distribution and contact area for attachments in each orthodontic treatment situation. (**a**) EX1; (**b**) EX2; (**c**) EX3; (**d**) EX4; and (**e**) EX5 for extrusion attachment; (**f**) IN1; (**g**) IN2; (**h**) IN3; (**i**) IN4; (**j**) IN5 for intrusion attachment; (**k**) RO1; (**l**) RO2; (**m**) RO3 for rotation attachment; (**n**) TO1; (**o**) TO2; (**p**) TO3; (**q**) TO4 for torque attachment.

**Figure 9 materials-13-03369-f009:**
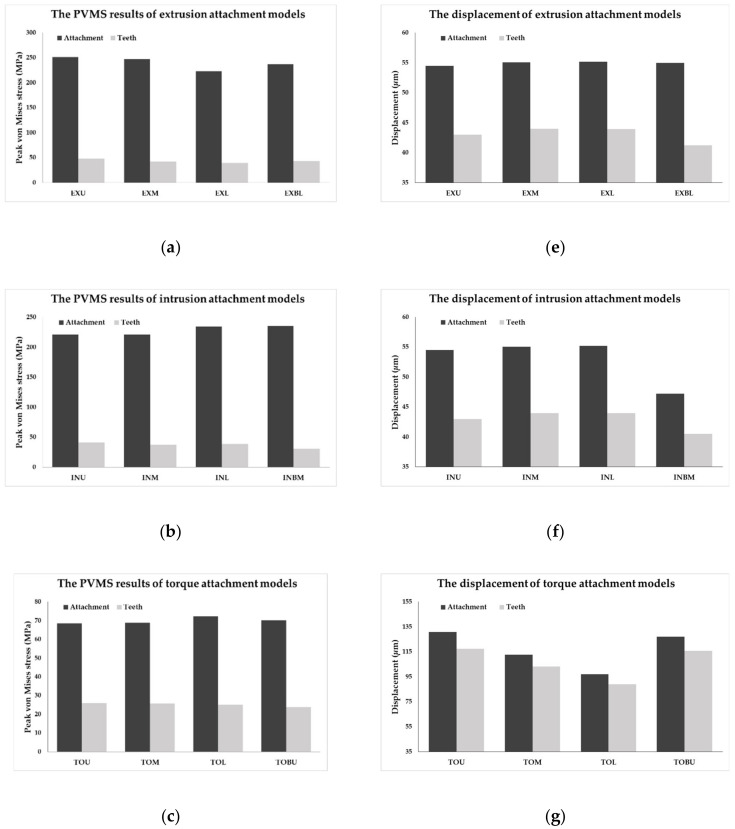
The PVMS values of (**a**) extrusion; (**b**) intrusion; (**c**) torque; and (**d**) rotation attachments; The displacement of (**e**) extrusion; (**f**) intrusion; (**g**) torque attachments; The rotation of (**h**) rotation attachment.

**Figure 10 materials-13-03369-f010:**
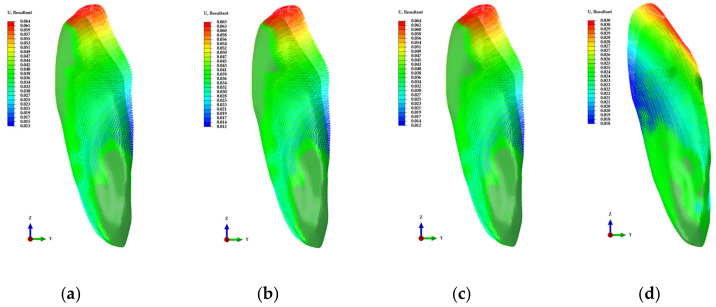
The displacement distribution of the teeth in various attachment models. (**a**) upper position; (**b**) middle position; (**c**) lower position in buccal side of the teeth; and (**d**) lower position in lingual side of the teeth for the extrusion attachment; (**e**) upper position; (**f**) middle position; (**g**) lower position in buccal side of the teeth; and (**h**) middle position in lingual side of the teeth for the intrusion attachment; (**i**) upper position; (**j**) middle position; (**k**) lower position in buccal side of the teeth; and (**l**) upper position in lingual side of the teeth for the torque attachment.

**Table 1 materials-13-03369-t001:** Material properties of the components in the finite element models.

Components	Young’s Modulus (MPa)	Poisson’s Ratio
Cortical bone [18,22]	13,700	0.3
Cancellous bone [18,22]	1370	0.3
Mucosa [22]	3.45	0.45
Teeth [23]	20,000	0.3
RTA [23]	2050	0.3
Attachment [19]	12,500	0.36

**Table 2 materials-13-03369-t002:** Parameters of the material property for the PDL.

Parameters	a_1_	u_1_	D_1_	a_2_	u_2_	D_2_
Value	2.5 × 10^−1^	5.5429 × 10^−3^	1.216 × 10^−1^	1.153 × 10^−1^	1.107 × 10^−1^	9.701 × 10^−1^

**Table 3 materials-13-03369-t003:** The Peak von Mises stress (PVMS) values (MPa) of orthodontic finite element analysis (FEA) models and contact pressure (MPa) of the attachment in four orthodontic treatment situations.

Types	Peak von Mises Stress (PVMS)					Contact Pressure
	Bone	Teeth	PDL	RTA	Attachment	Attachment
EX1	0.186	10.416	0.017	12.586	57.334	49.906
EX2	0.189	9.992	0.017	20.667	49.004	40.363
EX3	0.187	9.511	0.017	10.762	28.234	12.746
EX4	0.188	13.705	0.017	22.272	51.053	36.582
EX5	0.147	9.823	0.014	29.890	118.434	91.215
IN1	3.167	3.588	0.320	2.604	2.433	1.182
IN2	3.167	3.488	0.320	2.623	2.661	1.327
IN3	3.166	3.472	0.320	2.506	2.569	0.682
IN4	3.166	3.445	0.320	2.405	2.559	1.863
IN5	3.167	3.445	0.320	2.802	2.562	2.546
RO1	1.313	12.879	0.193	19.249	54.871	66.340
RO2	1.314	10.672	0.193	18.707	49.115	50.932
RO3	1.310	13.059	0.193	15.017	51.801	55.853
TO1	4.261	228.027	0.377	54.811	56.818	52.346
TO2	4.263	231.072	0.377	78.367	66.138	32.871
TO3	4.705	110.484	0.403	61.892	42.228	35.769
TO4	4.707	112.692	0.403	68.947	119.961	1515.150

**Table 4 materials-13-03369-t004:** Displacement and rotation values in the upper and lower regions, and contact area as well as contact rate in the attachment on the surface between the orthodontic device and attachment in four types of orthodontic treatment models.

Types	Teeth		Attachment		
	Upper Region	Lower Region	Total Area	Contact Area	Contact Rate
EX1	3.236 µm	3.068 µm	2.55 mm^2^	1.57 mm^2^	61.57%
EX2	3.273 µm	3.101 µm	2.63 mm^2^	2.05 mm^2^	77.95%
EX3	3.271 µm	3.075 µm	2.19 mm^2^	1.81 mm^2^	82.65%
EX4	3.267 µm	3.079 µm	2.05 mm^2^	1.15 mm^2^	56.10%
EX5	2.643 µm	2.492 µm	2.21 mm^2^	1.64 mm^2^	74.21%
IN1	48.887 µm	47.755 µm	2.55 mm^2^	1.34 mm^2^	52.55%
IN2	48.913 µm	47.750 µm	2.63 mm^2^	1.52 mm^2^	57.79%
IN3	48.890 µm	47.747 µm	2.19 mm^2^	1.46 mm^2^	66.67%
IN4	48.868 µm	47.744 µm	2.05 mm^2^	0.75 mm^2^	36.59%
IN5	48.884 µm	47.747 µm	2.21 mm^2^	1.15 mm^2^	52.04%
RO1	0.889 degrees	0.857 degrees	0.54 mm^2^	0.22 mm^2^	40.74%
RO2	0.890 degrees	0.858 degrees	0.52 mm^2^	0.21 mm^2^	40.38%
RO3	0.887 degrees	0.856 degrees	0.48 mm^2^	0.19 mm^2^	39.58%
TO1	59.442 µm	35.753 µm	1.83 mm^2^	0.52 mm^2^	28.42%
TO2	59.463 µm	35.764 µm	1.66 mm^2^	0.53 mm^2^	31.93%
TO3	58.114 µm	35.115 µm	1.54 mm^2^	0.72 mm^2^	46.75%
TO4	58.116 µm	35.111 µm	1.51 mm^2^	0.71 mm^2^	47.02%

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
