# Peer review of "Optimal Position of Attachment for Removable Thermoplastic Aligner on the Lower Canine Using Finite Element Analysis"

_materials, 2020, doi:10.3390/ma13153369_

Round 1
Reviewer 1 Report
Introduction
- “Orthodontic treatment is the diagnosis and treatment for correcting malocclusions such as open bite, cross bite, over bite, and crowding teeth [1-3].”
Given literature sources No. 1-3 do not correspond to the statement of the sentence at all. In addition, the spectrum of malocclusion is much greater than “open bite, cross bite, over bite, and crowding teeth”; and “over bite” is not a malocclusion, but one of the characteristics of occlusion. In general, the terminology used in the manuscript describing orthodontic treatment does not correspond to orthodontic terms.
- “Conventionally, this treatment method used arch wire, bracket, and interarch elastics which led to aesthetically dissatisfied patients and long treatment periods [4].”
The literature source does not correspond to the statement of the sentence. In addition, the spectrum of orthodontic appliances is much greater than “arch wire, bracket, and interarch elastics”.
- “In addition, the orthodontic devices manufactured during the treatment process are produced through manual analysis of the dental technicians, requiring a lot of time is required for proper movement of teeth during each orthodontic step [5].”
Please correct the sentence.
- “The advantages of RTA include excellent aesthetics, ease of wearing and removal, short chair time and reduced treatment duration.
Please motivate, why do you think that the treatment with RTA is shorter.
- “Previous studies have reported that root absorption does less occur in RTA as opposed to using brackets because the force is concentrated in the initial stage, giving the damaged cementum of the root the time to regenerate [7].”
The cited literature source is old and inappropriate. Also, not “root absorption”, but “root resorption”.
- “However, RTA shows a different orthodontic force compared to arch wire and brackets due to rebound force, making orthodontic treatment difficult in severe malocclusions because it cannot control the movement of the teeth as the surgeon intended [9-10].”
Why surgeon? Orthodontic treatment is planned and performed by an orthodontist.
- “Previous research has reported that attaching horizontal rectangular attachment with a thickness of 1 mm on the buccal and lingual sides helps the extrusion and rotation of teeth [12-13].
Both this sentence and the whole manuscript are based on very old literature, although clear aligners are a hot topic in orthodontics and there are many new articles.
Materials and Methods
- Please, explain, why Finite element analysis for deriving the optimum shape of attachments was performed on the simplified shapes of bone and teeth, while analysis of the attachment position was simulated using the mandibular CBCT image.
- Line 157: “ In addition, the teeth were divided into upper and lower areas with the PDL as the reference to measure the maximum displacement and rotation in each area. “
Please, provide the scheme. It is not clear where is the border- line between upper and lower areas.
- Line 180: “ For rotation attachments, a nine attachment positions were simulated by combining three anterior attachments and three posterior attachments attached to the anterior and to the posterior of the tooth, respectively”
In orthodontic terminology we use the terms “medial” and “distal”. Because it was not specified left or right side canine was used for analysis, it is more appropriate to use terms “medial” and “distal” rather than “anterior” and “posterior”. The same applies for the terms “ top” and “bottom” - the top is probably an “incisal position”, and the bottom is a “gingival position”?
- Figure 5- Confused ligual and buccal sides. As a result, I am not sure whether the sides are not confused in the whole study.
Results
- Line 249” “ In intrusion case, the EX3 showed the highest contact area (Figure 7f-j)….”- should not be IN3?
Discussion
Line 301: “ In addition, the use of RTA only may result in the movement that was not intended by the surgeon. “- Why surgeon? Orthodontic treatment is planned and performed by an orthodontist.
Conclusions
The conclusion “Therefore, it seems that using cylinder-shaped attachments at the posterior of the teeth is the most effective in orthodontic treatment using the removable thermoplastic aligner. “ - the conclusion seems too brave, because only separate tooth movements were analyzed, not the entire treatment.
Reviewer 2 Report
In the manuscript entitled: “Optimal Position of Attachment for Removable Thermoplastic Aligner on the Lower Canine using Finite Element Analysis” the authors identified the optimal shape and position of attachments by analyzing various shapes and positions of attachments.
The authors found that the optimum shape of cylinder was derived through the FEA of simple tooth shape and analyzing various positions of attachments on teeth revealed that fixing the attachments at the posterior of the tooth rather than the anterior allowed for torque control and an effective movement of the teeth
The authors concluded that desirable stress distribution was observed by inducing a high contact area between the attachment and the orthodontic device when a cylinder-shaped attachment was used on the removable thermoplastic aligner used in orthodontic treatment
Major comments:
In general, the idea and innovation of this study, regards the analysis of materials in orthodontics and aligners is interesting, because the analyses on these materials are validated but further studies on this topic could be an innovative issue in this field could be open an innovative matter of debate in literature by adding new information. Moreover, there are few reports in the literature that studied this interesting topic with this kind of study design.
The study was well conducted by the authors; However, there are some concerns to revise that are described below.
The introduction section resumes the existing knowledge regarding the important factor linked with biomaterials in orthodontic treatment.
However, as the importance of the topic, the reviewer strongly recommends, before a further re-evaluation of the manuscript, to update the literature through read, discuss and must cites in the references with great attention all of those recent interesting articles, that helps the authors to better introduce and discuss the aim of the study in light of a the impact of chewing muscles and tmj on the orthodontic forces: 1) Isola, G.; Anastasi, G.P.; Matarese, G.; Williams, R.C.; Cutroneo, G.; Bracco, P.; Piancino, M.G. Functional and molecular outcomes of the human masticatory muscles. Oral Dis. 2018, 24, 1428–1441, doi:10.1111/odi.12806.
The authors should be better specified, at the end of the introduction section, the rational of the study and the aim of the study with the null hypothesis. In the material and methods section, should better clarify how was performed the finite analysis and the treatment models. Moreover, specify if data were normalized or not. Please specify if was used a test unit.
The discussion section appears well organized with the relevant paper that support the conclusions, even if the authors should better discuss the importance of occlusal forces on biomaterials. The conclusion should reinforce in light of the discussions.
In conclusion, I am sure that the authors are fine clinicians who achieve very nice results with their adopted protocol. However, this study, in my view, does not in its current form, satisfy a very high scientific requirement for publication in this journal and requests a revision before a further re evaluation of the manuscript.
Minor Comments:
Abstract:
- Better formulate the introduction section by better describing the background
Introduction:
- Please refer to major comments
Discussion
- Please add a specific sentence that clarifies the results obtained in the first part of the discussion
- Page 13 last paragraph: Please reorganize this paragraph that is not clear
Round 2
Reviewer 2 Report
The authors have well addressed to both reviewers and editor comment. I suggest the acceptance of this interesting manuscript.